# A Brain-Inspired Hyperdimensional Computing Approach for Classifying Massive DNA Methylation Data of Cancer

**Fabio Cumbo** [1,*] **, Eleonora Cappelli** [2] **and Emanuel Weitschek** [3]

1   Department of Cellular, Computational and Integrative Biology (CIBIO), University of Trento, Via Sommarive 9, 38123 Povo Trento, Italy

2   Department of Engineering, University of Roma Tre, Via della Vasca Navale 79/81, 00146 Rome, Italy; eleonora.cappelli@uniroma3.it

3   Department of Engineering, Uninettuno University, Corso Vittorio Emanuele II 39, 00186 Rome, Italy; emanuel.weitschek@uninettunouniversity.net

*   Correspondence: fabio.cumbo@unitn.it

**Abstract:** The recent advancements in cancer genomics have put under the spotlight DNA methylation, a genetic modification that regulates the functioning of the genome and whose modifications have an important role in tumorigenesis and tumor-suppression. Because of the high dimensionality and the enormous amount of genomic data that are produced through the last advancements in Next Generation Sequencing, it is very challenging to effectively make use of DNA methylation data in diagnostics applications, e.g., in the identification of healthy vs diseased samples. Additionally, state-of-the-art techniques are not fast enough to rapidly produce reliable results or efficient in managing those massive amounts of data. For this reason, we propose HD-classifier, an in-memory cognitive-based hyperdimensional (HD) supervised machine learning algorithm for the classification of tumor vs non tumor samples through the analysis of their DNA Methylation data. The approach takes inspiration from how the human brain is able to remember and distinguish simple and complex concepts by adopting hypervectors and no single numerical values. Exactly as the brain works, this allows for encoding complex patterns, which makes the whole architecture robust to failures and mistakes also with noisy data. We design and develop an algorithm and a software tool that is able to perform supervised classification with the HD approach. We conduct experiments on three DNA methylation datasets of different types of cancer in order to prove the validity of our algorithm, i.e., Breast Invasive Carcinoma (BRCA), Kidney renal papillary cell carcinoma (KIRP), and Thyroid carcinoma (THCA). We obtain outstanding results in terms of accuracy and computational time with a low amount of computational resources. Furthermore, we validate our approach by comparing it (i) to BIGBIOCL, a software based on Random Forest for classifying big omics datasets in distributed computing environments, (ii) to Support Vector Machine (SVM), and (iii) to Decision Tree state-of-the-art classification methods. Finally, we freely release both the datasets and the software on GitHub.

**Keywords:** algorithms in biology; bioinformatics; machine learning; classification; hyperdimensional computing; cancer; DNA methylation

## 1. Introduction

In the last decade, the field of genomics has undergone great evolutions thanks to Next Generation Sequencing (NGS), a high-throughput technology for efficiently detecting the location, the structure, and the functionalities of the composing element (DNA) with low cost and high speed. NGS is adopted in cancer research with the aim to find novel biomarkers, factors of prediction, and targets for

achieving advancements in prognosis and treatments [1–4]. In particular, the NGS technique of DNA methylation [5,6] has been proven to play an important role in knowledge discovery of cancer [7–10]. Indeed, DNA methylation data are actually used in early diagnosis of tumors [11,12], because they enable finding potential biomarkers, i.e., specific patterns and characteristics that are common in tumor samples. DNA methylation is a biochemical modification that involves the addition of a methyl group in correspondence of carbon-5 in cytosine. This happens almost only in the dinucleotide CpG, which is the cytosine followed by a guanine [6]. This phenomenon is so common in DNA that it can be assessed that, among all of the CpG islands, 80% of them are methylated in mammals [13]. We can speak of hypo- or hyper-methylation, which regulate the turning on or off of genes that, for example, act as tumor-suppressors. Therefore, it becomes increasingly important to understand epigenetics modification, like DNA methylation, in human cells. In this work, we consider the data of DNA methylation experiments performed on cancer patients by focusing on three different types of cancer, i.e., Breast Invasive Carcinoma (BRCA), Kidney Renal Papillary Cell Carcinoma (KIRP), and Thyroid Carcinoma (THCA). These data are provided by the Genomic Data Commons (https://gdc.cancer.gov/) [14], a repository of cancer data, shared by several projects, such as The Cancer Genome Atlas (TCGA) [15], which we have considered. Indeed, it contains genomic and clinical data of more than 30 tumor types of over 20,000 patients derived from different NGS experiments, e.g., DNA methylation.

We design, develop, and apply a new machine learning algorithm, called HD-classifier, which relies on an in-memory cognitive-based hyperdimensional approach to distinguish tumor from non-tumor samples through the processing of their DNA methylation data in order to analyze those data. Machine learning techniques are very promising and wide adopted for the classification of cancer patients. Indeed machine learning (ML) has been used for cancer diagnosis and detection and also applied towards cancer prediction and prognosis [16]. ML is widely applied for case control studies, i.e., specific studies that aim to identify subjects by outcome status at the outset of the investigation, e.g., whether the subject is diagnosed with a disease. Subjects with such an outcome are categorized as cases. Once an outcome status is identified, controls (i.e., subjects without the outcome, but from the same source population) are selected. In supervised learning, a classification model is defined starting from labeled training data, with which the classifier tries to make predictions about unavailable or future data. Therefore, supervision means that, in the set of samples or datasets, the desired output signals (e.g., presence of a disease or not) are already known as previously labeled. In this type of learning, which relies on labels of discrete classes and on a training set, we will therefore have a task based on supervised techniques [10,12,17,18]. Classification problems are intended to identify the characteristics that indicate the group to which each sample belongs [19]. Previous studies that consider methylation data [10,20] have achieved good performance using state-of-the-art machine learning methods, e.g., rule based classifier [21] and decision trees [22,23], applied in order to produce classification models that are able to predict which class a new sample belongs. The final aim is to distinguish healthy and diseased samples in an effective way.

In this work, we propose HD-classifier, a brain-inspired Hyperdimensional (HD) computing algorithm that simulates the human brain by adopting (pseudo)random hypervectors, whose dimension is typically in the order of the thousands (usually $D = 10,000$). According to the HD paradigm [24,25], every atomic element is encoded into a HD vector that must be dissimilar to all the other vectors encoded in the same hyperspace. The fundamental idea of this computing paradigm is based on trying to emulate how the human brain works. In general, the methodology is able to remember and distinguish people, objects, and elements by working with patterns. With this logic, the HD computing is quite versatile and it can be applied to a wide range of scientific areas. For more details, we point the reader to [26], where the authors perform a comprehensive review of classification techniques based on HD computing. Indeed, HD computing was successfully applied in previous works on a limited set of research areas, like speech recognition [27,28], the internet of things [29–32], and two life-science related applications. These last attempts concern the detection of epileptogenic regions of the human brain from Intracranial

Electroencephalography (iEEG) recordings [33] and pattern matching problems on DNA sequences for diagnostic purposes [34,35]. Both of them produced highly promising results that led our research to extend the set of life-science applications to omics data. To our knowledge, our work constitutes the first application of a classification algorithm that is based on a brain inspired HD computing approach to DNA methylation data of cancer. Additionally, our software and algorithm can be easily used by the bioinformatics community also for other types of omics data, e.g., RNA sequencing [36].

## 2. Materials and Methods

Our experimental analysis is focused on public available data of The Cancer Genome Atlas (TCGA) [15] program hosted on the Genomic Data Commons (GDC) portal https://portal.gdc.cancer.gov/ [14]. This project aims to catalog the genetic mutations responsible for cancer, applying NGS techniques to improve the ability to diagnose, treat, and prevent cancer through a better understanding of the genetic basis of this disease. TCGA contains the genomic characterization and analysis of 33 types of tumors, and patient samples are processed through different types of NGS techniques, such as DNA sequencing, gene expression profiling, and DNA methylation. Additionally, TCGA collects and analyzes high quality cancer samples and makes the following data available to the research community: (i) clinical information on the patients participating in the program; (ii) samples metadata (for example, the weight of the sample, etc.); and, (iii) histopathological images of portions of the sample.

In this work, we focus on DNA methylation (DM) experiments, which describe the amount of methylated molecules for each of the known CpG regions on bisulfite-treated DNA sequences. If a methylation occurs on a DNA region, this causes the interruption of the transcriptional process of the promoter gene, where the involved region falls in. The amount of methylated molecules detected by the sequencing machine are represented by the beta value, [37]. Which is computed according to the Formula (1), where $Meth_n$ and $Unmeth_n$ are both intensities of the $n$-th methylated and unmethylated allele, respectively. The $\epsilon$ parameter acts as a modulator which is required when both the $Meth_n$ and $Unmeth_n$ are very low. The beta value is therefore a continuous variable in the (0, 1) range, where 1 represents full methylation and 0 no methylation at all.

$$\beta_n = \frac{max(Meth_n, 0)}{max(Meth_n, 0) + max(Unmeth_n, 0) + \epsilon} \tag{1}$$

The relevance of DNA methylation data has been widely discussed in the literature, and it has been proven that their processing can discriminate the diseased (tumor) from the healthy (also called normal) samples in case of cancer. For more details, we point the reader to Section 1.

The analysis of DNA methlyation data is challenging, because of the huge amount of CpG sites processed by the sequencing machines, i.e., up to 485,577 CpG sites, which represent the features in a supervised machine learning problem. Canonical classification approaches are not designed to manage hundred of thousands of features and most of the time fail in producing results, because of the required amount of computational resources. For this reason, we redesigned a classification procedure previously applied mainly on language detection and speech recognition problems. This method exploits high-dimensional numerical vectors (thus, hyperdimensional—HD) to represent information in the considered dataset. In particular, the atomic numerical values (or properly encoded categorical values) are represented as HD vectors, and the observations are also HD vectors built by combining their values—vector representations. In the context of DNA methylation data of cancer, we create two HD vectors representing the tumor and normal classes by collapsing all of the HD vector representations of the experiments considered during a first training phase. Thus, the class prediction of every sample is performed by computing the inner product between their HD representations and the classes HD vectors.

From a theoretical perspective, the whole procedure is as simple as powerful, and it can be summarised in Figure 1, and thoroughly explained with the steps that are described in the next subsections.

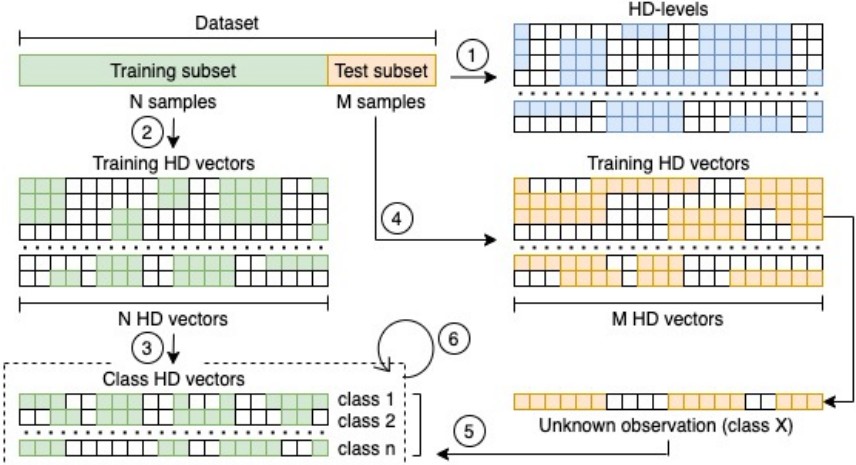

**Figure 1.** Flow chart of the whole procedure for building and testing the HD classification model. First, the definition of the set of HD-levels is required (1). It is a mandatory step toward the encoding of all the observations in the training subset (2) and the definition of the classes HD vector representations by combining the observations which belong to the same class (3). The same encoding procedure is then applied to the observations in the test subset (4). Every HD representation of the test observations is finally compared with the set of class HD vectors by computing the inner product (5). The class of the most closest vector is finally assigned to the observation. If a wrong class is assigned during the test phase, a retraining of the HD classification model must be performed multiple times (6) until the accuracy achieved in an iteration will not change as compared to the previous one.

### 2.1. Identifying HD-Levels

This is the first step that has to be performed before representing the data in the hyperdimensional space. The number of HD-levels is extremely relevant for maintaining the same precision of the numerical data as they are represented into the original dataset. In particular, consider a dataset that contains observations represented by numerical values ranging from 0.0 to 1.0 with a precision up to the third decimal number. As previously mentioned, the HD computing paradigm imposes that every atomic element in a dataset must be represented by a HD vector in the HD space. When considering our simple scenario with numerical values ranging from 0.0 up to 1.0 with a step of 0.001, a total number of 1,000 different randomly generated hypervectors are required in order to uniquely represent every numerical element in the hyperspace. These hypervectors are used to start encoding the dataset and build the classification model.

### 2.2. Encoding Data and Building the Classification Model

Encoding the features in a dataset is as important as choosing the amount of HD-levels. Additionally, for the numerical values representation of the observations, every feature must be encoded in the HD space by randomly generating unique hypervectors. Both the HD-levels and the feature hypervectors are essential to build the classification model, which is composed of a single hypervector for each class in a dataset and is built by collapsing all the HD representation of the classes observations. These last hypervectors are created by iterating over the values, one-by-one for each feature, and rotating the HD-level vector related to the current value by $n$ positions, where $n$ is the position of the feature among the whole list of features. Thus, the rotated HD-level vector must be summed element-wise to the HD representation of the feature. This procedure has been previously described in [27,28] with the name Ngram-based encoder. It consists of differentiating the feature positions through the assignment of unique hypervectors that result from the permutation of the feature HD representations, as described by Equation (2).

$$O_n^i = v_1 + \rho^1(v_2) + ... + \rho^{m-1}(v_m) \tag{2}$$

Equation (2) describes how a observation $n$, which belongs to the class $i$, is encoded in the HD space. The $\rho$ operator represents the rotation that must be applied $k$ times, with $k$ equaling to the position of the feature, while $v$ is the HD-level vector associated to a particular value. Finally, all of the resulting HD vector representations of the observations are summed together composing the HD representation of the class, as described with Equation (3).

$$C^i = O_1^i + O_2^i + ... + O_n^i \tag{3}$$

In this equation, $C^i$ is the HD representation of the class $i$, which is composed by applying the element-wise sum of all HD vector representations of the observations assigned to the class $i$. The same procedure is applied by grouping the observations assigned to different classes and can be generalised with the pseudocode in Algorithm 1.

---

**Algorithm 1** Encoding observations to class hypervectors

---

1: define *iter* as the maximum number of retraining iterations
2: *Dataset* is the input dataset
3: initialise *Classes* with the list of classes in *Dataset*
4: **for** *Class* in *Classes* **do**
5:     $vector \leftarrow [0 \ldots 0]$
6:     **for** *Obs* in *Class* **do**
7:         **for** $i = 0, 1, \ldots, m - 1$ in *Obs* **do**
8:             $l \leftarrow$ get the level vector according to the *Obs* value in the $i$-th position
9:             Rotate $l$ by $i$ positions
10:            Element-wise sum $l$ and *vector*
11:         **end for**
12:     **end for**
13:     yield *vector* as the HD representation of *Class*
14: **end for**
15: **for** $i = 0, ..., iter$ **do**
16:     retrain the classes vectors until the accuracy will not change
17: **end for**

---

By performing the element wise sum of the vectors, the final HD representation of a class will be very different from the single HD representation of the observations of that class. As described below, the classification process works on computing the distance between the vectors. For this reason, building the class hypervectors in this way could produce some wrong classification assignments and decrease the accuracy. Thus, the final class HD vectors must be fixed by performing a simple procedure, called retraining. After a HD representation of an observation is summed to the class HD vector, the distance between the observation and all the classes vectors is computed. If it results to be closest to the wrong class HD vector, then it must be element wise subtracted from all the classes vectors, except from the correct one. It must be finally summed again to the correct class HD vector. This procedure is repeated multiple times for all the observations until the ratio, which is defined as the number of wrongly classified observations on the total amount of observation, will remain stable between two consecutive iterations or will converge to zero.

*2.3. Class Prediction*

At this point, the classification model is ready to be tested. It is composed of the list of HD-levels and one hyperdimensional vector for each of the classes in the original dataset. A simple inner product between a testing observation and all the HD classes vectors must be performed in order to test the effective classification power of this model. The closest class according to this distance metric will be assigned to the observation vector. It is worth noting that the observation must be first encoded exploiting the same encoding procedure and the same set of HD-levels used for building the classification model.

## 3. Results

In order to prove the validity of our algorithm, we relied on the third party FTP repository, OpenGDC, which is a mirror of the GDC open accessible experimental and clinical data, all standardised in BED (Browser Extensible Data format) and extended with the integration of additional information retrieved from multiple data sources. The data in BED format are composed of 18 fields concerning information like the genomic coordinates of the CpG island, the name of the genes whose region overlap with the involved CpG island, etc. Two fields of the BED experimental data are relevant for our analysis, i.e., the *composite_element_ref* and *beta_value*, which contain the name/location of the CpG island and the measure of the amount of methylated molecules on the specific CpG island computed as previously described in Section 2.

We retrieved all of the DM experiments that were conducted on both normal and tumor tissues for Breast Invasive Carcinoma (BRCA), Kidney Renal Papillary Cell Carcinoma (KIRP), and Thyroid Carcinoma (THCA) tumors. Table 1 shows the number of samples and features for each of the datasets involved in our study.

**Table 1.** Compact overview of the datasets.

| Dataset | Tumor Samples | Normal Samples | Features |
|---------|---------------|----------------|----------|
| BRCA    | 799           | 98             | 485,512  |
| KIRP    | 276           | 45             | 485,512  |
| THCA    | 515           | 56             | 485,512  |

We created a matrix, where every row represents a sample, every column a CpG island (feature), and every element encodes the beta value (bv) of a particular region in a specific sample. Finally, we added an additional column that represents the class of the samples (i.e., tumor or normal) . An example of the final data matrix is illustrated in Table 2, where the number of columns (i.e., the features $m$) is equal to 485,577.

**Table 2.** Data matrix of DNA methylation data.

| Sample | CpG Site$_1$ | CpG Site$_2$ | $\cdots$ | CpG Site$_m$ | Class Label |
|--------|--------------|--------------|----------|--------------|-------------|
| $S_1$  | bv$_{11}$    | bv$_{12}$    | $\cdots$ | bv$_{1m}$    | normal      |
| $S_2$  | bv$_{21}$    | bv$_{22}$    | $\cdots$ | bv$_{2m}$    | tumor       |
| $\cdots$ | $\cdots$   | $\cdots$     | $\cdots$ | $\cdots$     | $\cdots$    |
| $S_n$  | bv$_{n1}$    | bv$_{n2}$    | $\cdots$ | bv$_{nm}$    | normal      |

For performing the supervised analysis, we randomly split the datasets by creating both the training and the test sub-matrices following the 80–20% proportion. We analysed the content of the matrices searching for the numerical precision used to represent the beta values. We found that these values are represented up to the 14th decimal number, which should result in defining $10^{14}$ HD levels in order to reflect the same precision in the hyperdimensional space. For obvious reason, maintaining $10^{14}$ hypervectors in memory requires a considerable amount of resources. Consequently, according to our analysis, we decided to use 1000 HD levels to made the whole process more efficient in terms of space and time by still maintaining a high precision, as demonstrated with the results that are highlighted below. We built a HD model for each of the datasets using 10,000 as the dimension of the hyperspace, and we finally test them with 100 retraining iterations. The results in terms of accuracy and both training and test running time are exposed in Table 3. Our experimentation with the HD classifier was performed on a consumer laptop with 8 GB of RAM and 1.2 GHz (Intel Core M) CPU using a single thread. To exclude any doubt concerning overfitting, we validated our experiments by applying the same procedure ten times on ten different randomly generated 80–20% training and test sets that were generated by random sampling the original datasets considered in our analyses. In Table 3, we depict the achieved benchmarks in terms of average training time, average classification time, and average accuracy.

**Table 3.** Performance of the HD classification algorithm.

| Dataset | Retraining Iterations | Avg. Training Time ± Std.Dev (h) | Avg. Classification Time ± Std.Dev (s) | Avg. Accuracy ± Std.Dev (%) |
|---------|----------------------|-------------------------------|--------------------------------------|----------------------------|
| BRCA | 4 | 5.44 ± 0.37 | 3 ± 0.68 | 97.7 ± 0.54 |
| KIRP | 1 | 2.97 ± 0.12 | 1 ± 0.32 | 98.4 ± 1.36 |
| THCA | 7 | 3.68 ± 0.47 | 2 ± 0.53 | 100 ± 0.002 |

Moreover, we also tried to reduce the amount of HD levels to check whether we were able to obtain comparable results by impacting on the memory required to store the levels dictionary. We noticed that by dynamically reducing the hyperdimensional space by 1000 up to 1000 we were able to still obtain great classification results in terms of accuracy for all three datasets. In particular, we also tried to cut the amount of HD-levels by varying them up to 600 with a step of 100. With the minimal amount of HD-levels and the lowest dimensionality of the HD space, we reached the 88.89%, 85.94%, and 90.35%, respectively, for BRCA, KIRP, and THCA, with a very high throughput in terms of speed and memory usage by still maintaining a high accuracy rate. We suggest the reader to refer to the Table 4 for a complete overview of our experimentation concerning the HD levels.

Table 4 shows that a low dimension of the HD vectors (1000) can produce an accuracy greater than 85% in all three datasets that are involved in our trials. For this reason, we further investigated how the accuracy of our HD classifier can be affected by fixing the size of the HD vectors to 1000 and varying the amount of HD-levels from 500 up to 1000 when considering a step of 100. It results in outstanding performances in terms of computational time and good performance in terms of accuracy with all the HD-levels configurations except for the lowest one of 500 that lowers the accuracy achievement for both the BRCA and KIRP datasets, as exposed in Table 5.

To validate the proposed approach, we compare our results with other supervised machine learning algorithms, one specifically designed for DNA methylation data and two other state-of-the-art ones.

The first experimentation is performed by applying BIGBIOCL [20] on the same datasets of this study. It is worth noting that BIGBIOCL is a supervised classification algorithm based of Random Forest, which has been designed to work specifically on omics data in distributed environments. Indeed, it is able to process datasets with a massive amount of features in few hours by extracting multiple alternative models by cutting out the features resulting significantly relevant in previous iterations. BIGBIOCL is able to run hundred of classification iterations in a very limited time window by distributing multiple jobs over the distributed computational environment. Additionally, in the case of BIGBIOCL, the results have been validated ten times with different randomly generated training (80%) ad test (20%) sets The results compared with the HD approach in terms of achieved accuracy (and standard deviations) are shown in Table 6. For what concerns the computational time, both the BIGBIOCL and HD approaches achieved comparable performances with in average 5.47 ± 0.37 and 4.21 ± 0.46 h for BRCA, 2.98 ± 0.12 and 2.13 ± 0.08 h for KIRP, and 3.70 ± 0.47 and 3.33 ± 0.13 for THCA, respectively.

**Table 4.** Benchmarks of the HD classification algorithm performances with different configurations of HD vectors dimensionality and HD-levels with a maximum amount of 100 retraining iterations. The best result achieved by the HD-classifier after the number of retraining iterations is shown. The configurations that led to a classification accuracy greater than 80% have been highlighted.

| Dataset | HD Vectors Dimensionality | HD-Levels | Retraining Iterations | Training Time (h) | Classification Time (s) | Accuracy (%) |
|---------|---------------------------|-----------|------------------------|-------------------|--------------------------|--------------|
| BRCA | 9000 | 900 | 100 | 10.53 | 6 | 56.11 |
|  | 9000 | 600 | 25 | 3.08 | 1 | 54.44 |
|  | 9000 | 500 | 100 | 3.07 | 6 | 57.78 |
|  | 7000 | 900 | 40 | 13.85 | 2 | 56.67 |
|  | 7000 | 700 | 52 | 5.08 | 3 | 57.78 |
|  | 7000 | 500 | 82 | 2.94 | 4 | 59.44 |
|  | 5000 | 900 | 5 | 3.51 | <1 | 51.11 |
|  | 5000 | 500 | 7 | 8.30 | <1 | 48.89 |
|  | 3000 | 900 | 84 | 2.79 | 3 | 58.33 |
|  | 3000 | 500 | 85 | 3.34 | 3 | 61.11 |
|  | **1000** | **900** | **3** | **2.33** | **<1** | **88.89** |
|  | 1000 | 500 | 65 | 11.41 | 2 | 60.00 |
| KIRP | 9000 | 900 | 94 | 3.67 | 2 | 64.06 |
|  | 9000 | 600 | 12 | 1.07 | <1 | 57.81 |
|  | 9000 | 500 | 29 | 1.05 | <1 | 59.38 |
|  | 7000 | 900 | 100 | 1.00 | 1 | 62.50 |
|  | 7000 | 700 | 69 | 1.00 | 1 | 62.50 |
|  | 7000 | 500 | 2 | 0.98 | <1 | 60.94 |
|  | 5000 | 900 | 25 | 0.94 | <1 | 64.06 |
|  | 5000 | 500 | 38 | 0.93 | <1 | 62.50 |
|  | 3000 | 900 | 94 | 0.82 | 1 | 62.50 |
|  | 3000 | 500 | 8 | 0.82 | <1 | 60.94 |
|  | **1000** | **900** | **3** | **0.78** | **<1** | **85.94** |
|  | 1000 | 500 | 24 | 0.77 | <1 | 59.38 |
| THCA | 9000 | 900 | 32 | 2.00 | 1 | 54.39 |
|  | 9000 | 600 | 18 | 2.10 | <1 | 57.89 |
|  | 9000 | 500 | 38 | 2.07 | 1 | 59.65 |
|  | 7000 | 900 | 100 | 1.91 | 3 | 64.04 |
|  | 7000 | 700 | 50 | 1.95 | 1 | 61.40 |
|  | 7000 | 500 | 43 | 2.39 | 1 | 61.40 |
|  | 5000 | 900 | 22 | 1.71 | <1 | 56.14 |
|  | 5000 | 500 | 26 | 1.69 | <1 | 56.14 |
|  | **3000** | **900** | **5** | **1.87** | **<1** | **100.0** |
|  | 3000 | 500 | 61 | 1.62 | 1 | 62.28 |
|  | **1000** | **900** | **3** | **1.47** | **<1** | **90.35** |
|  | **1000** | **500** | **6** | **4.95** | **<1** | **99.12** |

It is worth noting that BIGBIOCL has been executed on one of the 66 nodes of the PICO cluster of Cineca, which is the Italian inter-university consortium as well as the largest Italian computing centre, allocating 18 out of 128 GB of RAM and seven threads of a 2.5 GHz Intel Xeon E5 2670 v2 CPU (20 cores), whereas HD-classifier on a consumer laptop allocating just one thread of the 1.2 GHz (Intel Core M) CPU and 8GB of RAM.

Additionally, we compare our approach with a Decision Tree classifier (C4.5) and a Support Vector Machine (SVM), two state-of-the-art classification algorithms. In order to perform these comparisons, we applied the Weka software package [38] and tested the two classifiers on the BRCA dataset only with ten different randomly generated training (80%) and test (20%) sets. We summarised the achieved results with the SVM and the Decision Tree classifiers in Tables 7 and 8, respectively,( by varying their parameters.

**Table 5.** Benchmarks of the HD classification algorithm performances with different amounts of HD-levels and HD vector dimensionality fixed to 1000 and 100 maximum retraining iterations. The best result achieved by the HD-classifier after the number of retraining iterations is shown.

| Dataset | HD-Levels | Retraining Iterations | Training Time (h) | Classification Time (s) | Accuracy (%) |
|---------|-----------|----------------------|-------------------|------------------------|--------------|
| BRCA | 1000 | 2 | 2.30 | <1 | 88.89 |
| | 900 | 3 | 2.33 | <1 | 88.89 |
| | 800 | 3 | 6.77 | <1 | 88.89 |
| | 700 | 3 | 2.73 | <1 | 88.89 |
| | 600 | 3 | 4.76 | <1 | 88.89 |
| | 500 | 65 | 11.41 | 2 | 60.00 |
| KIRP | 1000 | 3 | 0.78 | <1 | 85.94 |
| | 900 | 3 | 0.78 | <1 | 85.94 |
| | 800 | 3 | 0.77 | <1 | 85.94 |
| | 700 | 3 | 0.77 | <1 | 85.94 |
| | 600 | 3 | 0.77 | <1 | 85.94 |
| | 500 | 24 | 0.77 | <1 | 59.38 |
| THCA | 1000 | 3 | 1.38 | <1 | 90.35 |
| | 900 | 3 | 1.47 | <1 | 90.35 |
| | 800 | 3 | 1.37 | <1 | 90.35 |
| | 700 | 2 | 1.38 | <1 | 90.35 |
| | 600 | 2 | 1.35 | <1 | 90.35 |
| | 500 | 6 | 4.95 | <1 | 99.12 |

**Table 6.** Comparison of HD-classifier vs. BIGBIOCL classifier on ten different randomly generated training (80%) and test (20%) sets for each tumor dataset.

| Dataset | HD-Classifier Avg. Accuracy $\pm$ Std.Dev (%) | BIGBIOCL Avg. Accuracy $\pm$ Std.Dev (%) |
|---------|-----------------------------------------------|------------------------------------------|
| BRCA | 97.7 $\pm$ 0.54 | 98.0 $\pm$ 0.94 |
| KIRP | 98.4 $\pm$ 1.36 | 97.0 $\pm$ 2.04 |
| THCA | 100 $\pm$ 0.002 | 97.0 $\pm$ 0.03 |

It is worth noting that all the proposed comparisons have been performed in multithreading. We tried to also repeat these experiments by running the sequential implementation of the classification algorithms, with no results, even after days of computation. It is also worth to note that we were able to perform our analyses with the HD classifier on a consumer laptop with 1.2 GHz (Intel Core M) CPU and 8 GB of RAM in single thread, proving that the HD approach can handle massive datasets with a very limited amount of computational resources.

**Table 7.** SVM classifier performance on the Breast Invasive Carcinoma (BRCA) dataset with ten different randomly generated training (80%) and test (20%) sets.

| Memory | Threads | Regularization Method | Regularization Parameter | Iterations | Avg. Execution Time (h) | Avg. Accuracy $\pm$ Std.Dev (%) |
|--------|---------|----------------------|--------------------------|------------|-------------------------|--------------------------------|
| 12 GB | 7 | L2 | 1.0 | 100 | 2.05 $\pm$ 0.11 | 98.95 $\pm$ 1.54 |
| 12 GB | 7 | L2 | 1.0 | 200 | 3.53 $\pm$ 0.02 | 98.97 $\pm$ 2.04 |
| 12 GB | 7 | L1 | 0.1 | 100 | 1.67 $\pm$ 0.07 | 95.46 $\pm$ 0.63 |
| 12 GB | 7 | L1 | 0.1 | 200 | 1.32 $\pm$ 0.04 | 99.16 $\pm$ 1.35 |

**Table 8.** Decision Tree classifier performance on the BRCA dataset with ten different randomly generated training (80%) and test (20%) sets.

| Memory | Threads | Max Depth | Max Bins | Impurity | Avg. Execution Time $\pm$ Std.Dev (min) | Avg. Accuracy $\pm$ Std.Dev |
|--------|---------|-----------|----------|----------|----------------------------------------|-----------------------------|
| 5 GB   | 4       | 5         | 16       | Gini     | $55.2 \pm 3.02$                        | $98.51 \pm 1.34$            |
| 5 GB   | 4       | 5         | 32       | Gini     | Out of Memory                          | -                           |
| 12 GB  | 7       | 5         | 32       | Gini     | $68.19 \pm 2.74$                       | $98.76 \pm 0.83$            |
| 12 GB  | 7       | 10        | 32       | Gini     | $69.88 \pm 3.68$                       | $99.20 \pm 0.38$            |
| 12 GB  | 7       | 5         | 8        | Gini     | $11.52 \pm 0.86$                       | $98.03 \pm 2.17$            |
| 18 GB  | 7       | 5         | 128      | Gini     | Out of Memory                          | -                           |

## 4. Discussion, Conclusions, and Future Directions

Our approach for classifying tumor vs non tumor samples through their DNA methylation data is proposed as a starting point for the application of HD computing to the life science domain. Building a HD model is much easier than any other procedure involving state-of-the-art classification algorithms, but we demonstrated that it can reach and overcome them in terms of accuracy, running time, and memory usage. We want to highlight that our experiments have not been performed in parallel. However, the proposed approach can be easily run in parallel on both CPU and GPU architectures due to the nature of the HD paradigm [39,40]. This would drastically speed up the procedure for building the classification model and the prediction of a class for new observations. Moreover, because the HD-classifier is very sensitive and depends on the number of HD-levels used to encode the data, a fine tuning procedure of the method is sometimes necessary in order to produce the best results in term of accuracy. Indeed, we are currently working on a new method for the automatic estimation of the best parameters for the HD algorithm in order to achieve the best performances in terms of computational time, memory consumption, and accuracy. Additionally, our proposal can work on pure classification tasks only, without extracting relevant features, as some supervised machine learning algorithms can perform, e.g., rule- or tree-based classification algorithms. However, this functionality can be possibly defined by analysing the content distribution of the HD representations of the classes and identifying the features that are involved in the definition of patterns that can discriminate one class from the others. These features correspond to specific CpG islands that are overlapped to the genomic region of known genes. If the identified genes are known in literature to be responsible for the advancement of a specific disease, our proposal could be adopted for diagnostic purposes.

To summarize, we presented a brain-inspired hyperdimensional classifier. The proposed approach has been applied on three biological datasets concerning experiments that were conducted on both healthy and diseased individuals affected by three different types tumors, e.g., Breast Invasive Carcinoma (BRCA), Kidney Renal Papillary Cell Carcinoma (KIRP), and Thyroid Carcinoma (THCA). We validated our method by comparing the achieved results with three different classification algorithms, one specifically designed for omics data ( BIGBIOCL) and two state-of-the-art ones (Decision Trees and Support Vector Machines). Our approach overcomes all of them in terms of computational time, the amount of memory, and achieved accuracy, highlighting that it efficiently manages massive datasets on consumer laptops with a very limited amount of computational resources. Therefore, HD computing is a valid highly efficient alternative to state-of-the-art classification algorithms. The very limited set of arithmetic operations required to encode a dataset, i.e., element wise sum and subtraction of vectors, makes the HD-classifier the first candidate for the novel emergent class of processors, which has the main goal of consuming just a small fraction of the energy actually consumed from the current class of CPUs. Additionally, the promising results achieved in terms of speed and accuracy make our approach a good candidate for quasi-real-time analytics applications in healthcare and life sciences.

In the future, we plan to extend the experimentation on other omics datasets, in order to confirm the validity of our approach. Additionally, we want to improve the computational performance of

our classifier by implementing a multi-threading version of our algorithm through CPUs and GPUs computing. Finally, we are investigating the possibility to use the vector representations of HD computing to improve the feature selection step in supervised and unsupervised machine learning problems.

**Author Contributions:** F.C. and E.C. collected the datasets. F.C. and E.W. performed the analyses. F.C. and E.W. conceived and directed the research. E.W. supervised the research. All authors have read and agreed to the published version of the manuscript.

**Funding:** This research received no external funding.

**Acknowledgments:** We wish to thank Simone Truglia, a post-graduate student of the Master Degree program in Computer Science of the Department of Engineering, Uninettuno University for the experimentation about the different HD configurations.

**Conflicts of Interest:** The authors declare no conflict of interest.

## Abbreviations

The following abbreviations are used in this manuscript:

| | |
|---|---|
| BRCA | Breast Invasive Carcinoma |
| DM | DNA methylation |
| DT | Decision Tree |
| GDC | Genomic Data Commons |
| HD | Hyperdimensional |
| KIRP | Kidney renal papillary cell carcinoma |
| ML | Machine Learning |
| SVM | Support Vector Machine |
| TCGA | The Cancer Genome Atlas |
| THCA | Thyroid carcinoma |

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
