# Peer review of "A Brain-Inspired Hyperdimensional Computing Approach for Classifying Massive DNA Methylation Data of Cancer"

_algorithms, doi:10.3390/a13090233_

Round 1
Reviewer 1 Report
The paper "A brain-inspired hyperdimensional computing approach for classifying massive DNA methylation data of cancer" presents an application of a machine learning classification method to the problem of distinguishing between normal and tumor samples based on methylation data. Overall, based on the results presented in the paper, I can conclude that there is some potential of applying this method in practice, however, the results presented in the paper do not provide an evidence that it's a clear winner. By the way, why does the method not have a name? It should not be just "HD method".
I have some objections to the paper that I will elaborate on further in this review.
General thoughts
1) Although the paper is self-contained in the sense that the reader can read it without consulting many of the external resources (for example, other research papers that are referenced), its exposition sometimes seems to be too redundant. For example, it seems to me that the introduction is quite wordy - starting it out with talking about the fact that the DNA is composed of A-C-T-G is verbose. I would start the introduction with 1-2 sentences explaining what methylation is and what is its importance and then proceed to talk more about the algorithmic challenges and the existing work.
2) Materials and Methods - again, too wordy - why should the reader know the fact that on July 15th 2016 TCGA portal was officially closed? To me, it has no relevance to the paper.
3) English has to be improved substantially, however, the reader is able to understand the work with almost no effort. I will not point out every single grammar error that I spotted - the authors are recommended to make a couple of thorough passes through the text.
Concrete comments
1) I did not understand why the algorithm presented in the paper works well. What is its main advantage? Is this a completely novel approach? There are only two references to Ngram-based encoders and no references or explanation of the algorithm main idea. Also, does it provably converge or it is a heuristic approach? Since this is an algorithmic paper in an algorithmic journal, more details must be provided.
2) I am worried about your results. For example, when I look at Table 6, I am not convinced that I would use HD versus BigBioCl - its runtime is larger and the difference between the results is not significant. Is it possible to have 100.0 accuracy (THCA)? It is something fantastic.
3) Table 5 suggests a very weak spot of the HD method - it is very sensitive depending on the number of HD-levels used to encode the data. For example, in BRCA, 500 levels yield the worst result and in THCA 500 levels provide the best result. This issue should at least be raised in the discussions - that's a serious drawback. Additionally, a guidance (if any) on how to choose the number of HD levels should be provided.
Author Response
Answers to Reviewer 1
The paper "A brain-inspired hyperdimensional computing approach for classifying massive DNA methylation data of cancer" presents an application of a machine learning classification method to the problem of distinguishing between normal and tumor samples based on methylation data. Overall, based on the results presented in the paper, I can conclude that there is some potential of applying this method in practice, however, the results presented in the paper do not provide an evidence that it's a clear winner.
We thank the reviewer for the positive comments and for the suggested improvements that we addressed during this revision process.
By the way, why does the method not have a name? It should not be just "HD method".
Thank you for your suggestion, we called our algorithm and software HD-classifier
I have some objections to the paper that I will elaborate on further in this review.
Please find our point-to-point answers below.
General thoughts:
Although the paper is self-contained in the sense that the reader can read it without consulting many of the external resources (for example, other research papers that are referenced), its exposition sometimes seems to be too redundant. For example, it seems to me that the introduction is quite wordy - starting it out with talking about the fact that the DNA is composed of A-C-T-G is verbose. I would start the introduction with 1-2 sentences explaining what methylation is and what is its importance and then proceed to talk more about the algorithmic challenges and the existing work.
We shortened the introduction by removing basic definitions and wordy parts. Additionally, we reformulated some phrases and removed the sentences in lines 30-35 and 37-43.
Materials and Methods - again, too wordy - why should the reader know the fact that on July 15th 2016 TCGA portal was officially closed? To me, it has no relevance to the paper.
Sentences from line 110 to 142 have been removed and the section has been improved.
English has to be improved substantially, however, the reader is able to understand the work with almost no effort. I will not point out every single grammar error that I spotted - the authors are recommended to make a couple of thorough passes through the text
We accurately revised grammar, terminology, and language for improving the overall English level.
Concrete comments:
I did not understand why the algorithm presented in the paper works well. What is its main advantage? Is this a completely novel approach? There are only two references to Ngram-based encoders and no references or explanation of the algorithm main idea. Also, does it probably converge or it is a heuristic approach? Since this is an algorithmic paper in an algorithmic journal, more details must be provided.
The proposed approach is based on the HD computing principles (see references [28-39] and [43]). The main idea started with a research conducted by Pentti Kanerva on 2009 of the Redwood Neuroscience Institute who defined a novel computing paradigm based on representing everything with vectors in high dimensions. It works pretty well because Kanerva was able to identify how the human brain works in remembering and distinguishing things. A keyboard is different from a mouse, but there are a lot of different types of keyboards, with different shapes, colours, etc. The HD representation of all the possible types of keyboards are very similar (but not equals) and thus they belong to the same conceptual class. However, they have some small differences in their shapes. These differences characterise their particular details (shape, colour, etc. in this example). From this simple idea, a lot of applications were born during the last decade which all use the same principles ([28-39], [43]).
In section methods and its subsections we explain our algorithm and detail all the computational steps. In particular, figure 1, equations 2,3, and algorithm 1 provide a schematic overview of the method. During this revision we further improved the description of the method, software, and algorithm.
I am worried about your results. For example, when I look at Table 6, I am not convinced that I would use HD versus BigBioCl - its runtime is larger and the difference between the results is not significant. Is it possible to have 100.0 accuracy (THCA)? It is something fantastic.
BIGBIOCL was executed on a cluster for big data computing with a 2.5GHz Intel Xeon E5 2670 v2 CPU allocating 7 threads and 18 GB of RAM, whereas HD-classifier on a consumer laptop with a 1.2GHz Intel CPU, single thread, and 8Gb of RAM. We inserted and highlighted this fact in lines 264-269 of page 8. Additionally, we moved the information about the computational time for both the BIGBIOCL and HD approaches outside Table 6. BIGBIOCL has been designed to work on massive omics datasets and it exploits big data technologies to analyze them in distributed environments. On the other hand, we demonstrated that the HD method is able to work pretty well on massive omics datasets by using just a fraction of the computational resources required to run BIGBIOCL. Additionally, it is worth noting that the algorithm was able to achieve the 100% accuracy on the THCA dataset after 7 retraining iterations (94.73%, 96.49%, 98.24%, 98.24%, 98.24%, 99.12%, and finally 100%, we also rounded the average accuracy of 99.99 obtained on ten different randomly generated training and test sets). The main goal of the retraining procedure is to smooth the shape of the classes hypervectors and make them as different as possible by subtracting the HD vector representations of the observations which belong to a class from all the HD vectors of the other classes. The retraining procedure has been widely discussed in Mohsen Imani, et al. "Hierarchical hyperdimensional computing for energy efficient classification." 2018 55th ACM/ESDA/IEEE Design Automation Conference (DAC). IEEE, 2018 also cited in our manuscript.
Table 5 suggests a very weak spot of the HD method - it is very sensitive depending on the number of HD-levels used to encode the data. For example, in BRCA, 500 levels yield the worst result and in THCA 500 levels provide the best result. This issue should at least be raised in the discussions - that's a serious drawback. Additionally, a guidance (if any) on how to choose the number of HD levels should be provided.
The results presented with Table 4 and Table 5 show that the performances of the HD method are sensitive to the variations of the number of HD levels. Looking at the content and the distribution of the values in the original matrix is a good practice that helps decide the best number of HD levels, as described in the section “Identifying HD levels”. However, we are currently working on a new method for the automatic estimation of the best parameters for the HD algorithm in order to achieve the best performances in terms of computational time, memory consumption, and accuracy. We added such considerations on pages 10-11 lines 288-292.
Reviewer 2 Report
The authors presented a new approach to solve the classification problem of DNA methylation in cancer data. Method is quite new, does not require a substantial computational resources and is characterised by high accuracy. Despite these positives features I have some critical remarks of the article listed below.
Major remarks:
- The beginning of Section 2. Do you really think that a description of GDC in such details is needed. In my opinion it is out of scope.
- One thing is unclear for me. In the section “Identifying HD-levels”, in example we had a precision up to 3rd decimal number and therefore we had 10^3 HD levels. On page 7, line 220, we had values represented up to 14th decimal number, and we had 14000 HD levels not 10^14? On the other hand, the values in the table are beta values, so I assume that so many potentially different values are (or could be) undistinguishable because (probably) the estimation of the methylation level is burdened with higher error.
- As it is presented on Figure 1, in classifier there can be more than two classes. Therefore why did not you put all tested cancer samples belongs to the different classes into one test set? It would be very interesting to see the accuracy of the method during determining only between tumour-normal classes, but BRCA/KIRP/THCA/normal classes. I understand that from practical point of view, the clinicians would like to get an information “specific tumour/normal” about the sample, because of their additional knowledge about it, but I suppose that some genetic markers are common for many different type of cancer and probably it would be harder to detect also the type of cancer. Moreover, what would be the answer of the classifier built for example for BRCA cancer, when KIRP sample will be applied to the input?
- Results of test presented in Table 3 should be clearly explained. The authors mentioned that they applied 100 times the procedure to different 80%-20% training and testing generated matrices by random sampling. So, the time is a sum for these 100 matrices? Or average? If average, what about standard deviation. I do not believe that all 100 times the computational time was exactly the same (since it depends on number of retraining iterations). On the other hand, I think that if someone would like to use the method in a production pipeline, the training time can be neglected. Even if it took a year, the algorithm would be worth attention with high accuracy. So, why when comparing to other ML solutions, the training and classifying times are not given (I mean results in tables 6, 7, 8 – see also p. 7 below).
- Comparing calculation times for algorithms run on different platforms has no big sense and therefore presenting processing times (table 6) without precise information that the calculation was performed on different platforms can be confusing. I suggest to remove the two columns from table 6 and left only description in text or highly expand the description of table 6. Obviously, the information is important – similar time for computationally limited laptop and multithread cluster. But do not put it in one table without description, please.
- Table 4 and 5 – classification time 0 looks awkward. I suggest to change it into “below 1s”. I could take a bet that it took some time, anyway.
- Can we compare “training time” (table 5) with “build time” (table 8) and “classification time” (table 5) with “evaluation time” (table 8)? What about “execution time” (table 7)? It is important in context a sentence (page 11, line 305) that the presented solution can be used in quasi-real-time analytics application. Probably, other ML solution also as far as its classification time is short. (see my remark 4 above about neglecting the training time). On the basis of the presented results we cannot say that the presented method outperforms other solution from the classification accuracy point of view.
- Do the test presented in Table 4 and Table 5 was also performed 100 times? If not the results can be confusing, because they are based on a single draw and it can be a dumb luck. If they was performed 100 times are the results presents sum, avg, and what about a standard deviation of results?
Minor remarks:
- I have a strange filling that the authors could not decide to whom their article is addressed. For example, Page 2, line 32 where the information about 4 letters DNA is presented contrasts with very sketchy description of DNA methylation. I suggest to discard the sentence about DNA sequencing.
- “Tumoral samples” or “tumor samples”? I am not sure about this, but “tumor” or “tumour” is much more often used.
- Please consider to use a consistent name: “retraining” or “re-training”.
- Page 2, line 48. The sentence “This phenomenon is so common in DNA that it can be assessed that among all the CpG islands, the 80% of all them is methylated in mammals.” sounds awkward. Does it mean that the rest 20% is methylated in non-mammals? Moreover, “all” after “80% of” should be removed and.
- Page 3, line 80. I suggest to avoid the words “energy efficient” as it concerns the examination of the efficiency of algorithms in terms of energy consumption of the platform on which the calculations are performed.
- Page 3, line 97. The sentence “… our software and algorithm can be successfully used by the bioinformatics community also for other types of omnics data.” Is a speculation. How do you know, that it can be “successfully” used? J
- Page 4, lines without numbers: in fact you described beta value two times.
- Page 4, lines without numbers: the “nth” should looks probably like “n-th” not “n to the power of t, multiplied by h”.
- Page 4, lines 143-145. The sentence means exactly the same as described above in unnumbered paragraph.
- Page 5, description of Figure 1 – the number 6, which is marked on the Figure, is missing in the description.
- Page 5, line 170. Number not amount – amount is for uncountable nouns.
- Page 6 (not only) Algorithm 1. I strongly suggest to use the same letters for the number of samples, features etc. in whole article. For example – Algorithm 1 – M means number of features then, Table 2 (page 7) – the number of features is “m”. Number of samples is “n” or “N”. Take a look on your equations, Figure 1, Algorithm 1 and tables and use the same letters, please.
- Page 7, I suggest to read again the first three sentences of section 3. In my opinion the information there are doubled – i.e. the “definition” of OpenGDC.
- Page 7, line 209. The section number at the end of the sentence is missing.
- Page 7, line 212. Not “amount” of samples but the “number” of samples. Amount is for uncountable nouns.
- Page 7, line 215. What for is “i” at the end of the sentence? Should it be italic?
- BigBioCl or BIGBIOCL or BIGBIOGL? All of them are in the article, but only one is correct.
Author Response
Answers to Reviewer 2
The authors presented a new approach to solve the classification problem of DNA methylation in cancer data. Method is quite new, does not require a substantial computational resources and is characterised by high accuracy. Despite these positives features I have some critical remarks of the article listed below.
We thank for the positive comments and for the precious suggestions that helped us to improve our work. A point-to-point answer to the raised issues is provided below.
Major remarks:
The beginning of Section 2. Do you really think that a description of GDC in such details is needed. In my opinion it is out of scope.
We removed most of the content which described the GDC portal (lines 110-142). However, we kept a brief paragraph (lines 100-109) in order to properly introduce the source from which we extracted the DNA methylation data used in our experiments.
One thing is unclear for me. In the section “Identifying HD-levels”, in example we had a precision up to 3rd decimal number and therefore we had 10^3 HD levels. On page 7, line 220, we had values represented up to 14th decimal number, and we had 14000 HD levels not 10^14? On the other hand, the values in the table are beta values, so I assume that so many potentially different values are (or could be) undistinguishable because (probably) the estimation of the methylation level is burdened with higher error.
This was a mistake. 10^14 is the theoretical number of required hypervectors to reflect the same numerical accuracy of the dataset. We corrected the wrong information and explained better this fact (lines 221-224).
As it is presented on Figure 1, in classifier there can be more than two classes. Therefore why did not you put all tested cancer samples belongs to the different classes into one test set? It would be very interesting to see the accuracy of the method during determining only between tumour-normal classes, but BRCA/KIRP/THCA/normal classes. I understand that from practical point of view, the clinicians would like to get an information “specific tumour/normal” about the sample, because of their additional knowledge about it, but I suppose that some genetic markers are common for many different type of cancer and probably it would be harder to detect also the type of cancer. Moreover, what would be the answer of the classifier built for example for BRCA cancer, when KIRP sample will be applied to the input?
We created three classification models, one for each dataset. The observation about merging BRCA, KIRP, and THCA is theoretically possible because the involved CpG islands are always the same among the datasets. However, the experiments have been conducted on biological samples that come from different anatomical regions (breast, kidney, and thyroid in the case of BRCA, KIRP, and THCA respectively). The reviewer suggest to use a non-binary number of classes (BRCA/KIRP/THCA/normal) but this requires to consider the normal samples of a tumor dataset on the same level of the normal samples of the other tumor datasets. Biologically speaking, this is not correct, because the methylation process of the DNA could occur on specific CpG islands in the kidney renal cells but completely different DNA regions could be involved in the methylation of the DNA in the thyroid cells (same consideration for any kind of biological samples).
Results of test presented in Table 3 should be clearly explained. The authors mentioned that they applied 100 times the procedure to different 80%-20% training and testing generated matrices by random sampling. So, the time is a sum for these 100 matrices? Or average? If average, what about standard deviation. I do not believe that all 100 times the computational time was exactly the same (since it depends on number of retraining iterations). On the other hand, I think that if someone would like to use the method in a production pipeline, the training time can be neglected. Even if it took a year, the algorithm would be worth attention with high accuracy. So, why when comparing to other ML solutions, the training and classifying times are not given (I mean results in tables 6, 7, 8 – see also p. 7 below).
We applied 100 retraining iterations and ten different 80%-20% randomly generated training and test sets. We clarified this fact in the manuscript (page 8 lines 232). The times and accuracies are averages.We added the standard deviations to table 3 and to the other tables when reporting the comparisons. We also uniformed the names of build time, evaluation time, which were mentioned in table 8. Additionally, we moved the computational times for both the HD and BIGBIOCL approaches outside the Table 6 as requested with the comment n. 5, while in Table 8 the columns “Build Time” and “Evaluation Time” have been renamed to “Training Time” and “Classification Time” respectively (see our answer to the comment n. 7).
Comparing calculation times for algorithms run on different platforms has no big sense and therefore presenting processing times (table 6) without precise information that the calculation was performed on different platforms can be confusing. I suggest to remove the two columns from table 6 and left only description in text or highly expand the description of table 6. Obviously, the information is important – similar time for computationally limited laptop and multithread cluster. But do not put it in one table without description, please.
BIGBIOCL was executed on a cluster for big data computing with a 2.5GHz Intel Xeon E5 2670 v2 CPU allocating 7 threads and 18 GB of RAM, whereas HD-classifier on a consumer laptop with a 1.2GHz Intel CPU, single thread, and 8Gb of RAM. We highlighted this fact in lines 264-269 of page 8. As suggested, we moved the information about the computational time for both the BIGBIOCL and HD approaches outside Table 6.
Table 4 and 5 – classification time 0 looks awkward. I suggest to change it into “below 1s”. I could take a bet that it took some time, anyway.
We fixed Table 4 and Table 5 by replacing zeros under the “Training time (seconds)” and “Classification time (seconds)” with “< 1”. We additionally added a tilde before the time values;
Can we compare “training time” (table 5) with “build time” (table 8) and “classification time” (table 5) with “evaluation time” (table 8)? What about “execution time” (table 7)? It is important in context a sentence (page 11, line 305) that the presented solution can be used in quasi-real-time analytics application. Probably, other ML solution also as far as its classification time is short. (see my remark 4 above about neglecting the training time). On the basis of the presented results we cannot say that the presented method outperforms other solution from the classification accuracy point of view.
Yes, the “training time” is the same as “build time”, and “classification time” is the same as “evaluation time”. We used the expression “execution time” to indicate the whole time required by the software to complete its execution. We uniformed the terminology used in the manuscript. Thanks to the reviewer for reporting this issue. Additionally, we would like to let the reviewer note that our approach is comparable with those one used for our testing in terms of achieved accuracy, but it outperforms them in terms of required computation time and resources, as highlighted in section Results.
Do the test presented in Table 4 and Table 5 was also performed 100 times? If not the results can be confusing, because they are based on a single draw and it can be a dumb luck. If they was performed 100 times are the results presents sum, avg, and what about a standard deviation of results?
The test presented in Table 4 and Table 5 have been performed with 100 retraining iterations on ten different randomly generated training (80%) and test (20%) sets. It is worth noting that after a retraining iteration, the software could drop or grow its accuracy with respect to the previous iteration. In Table 4 and Table 5 we show the best result achieved by the HD-classifier after the number of retraining iterations (see column “Retraining iterations”). It is also worth noting that in some cases our approach is not able to obtain high accuracy levels even after 100 retraining processes (see the first, fourth, and fourth trials for BRCA, KIRP, and THCA respectively in Table 4).
Minor remarks:
I have a strange filling that the authors could not decide to whom their article is addressed. For example, Page 2, line 32 where the information about 4 letters DNA is presented contrasts with very sketchy description of DNA methylation. I suggest to discard the sentence about DNA sequencing.
We improved the introduction and removed wordy phrases. Indeed, we removed the sentence about the DNA sequencing on line 30-32.
“Tumoral samples” or “tumor samples”? I am not sure about this, but “tumor” or “tumour” is much more often used.
We uniformed the text by changing the “tumoral” word with the “tumor”.
Please consider to use a consistent name: “retraining” or “re-training”.
We also uniformed the text by changing “re-training” with “retraining”.
Page 2, line 48. The sentence “This phenomenon is so common in DNA that it can be assessed that among all the CpG islands, the 80% of all them is methylated in mammals.” sounds awkward. Does it mean that the rest 20% is methylated in non-mammals? Moreover, “all” after “80% of” should be removed and.
It is known in literature that around 80% of the whole set of CpG islands are methylated in mammals. We highlighted this because we worked with human samples and we considered this a relevant information to be noted especially for those readers with a limited biological background. However, this does not say anything about non-mammals, whose methylation level on their CpG sites may vary depending on the species.
Page 3, line 80. I suggest to avoid the words “energy efficient” as it concerns the examination of the efficiency of algorithms in terms of energy consumption of the platform on which the calculations are performed.
We removed “energy efficient” from line 80.
Page 3, line 97. The sentence “… our software and algorithm can be successfully used by the bioinformatics community also for other types of omnics data.” Is a speculation. How do you know, that it can be “successfully” used?
We changed the term “successfully” to “easily” in line 97.
Page 4, lines without numbers: in fact you described beta value two times.
We shortened the definition and removed redundant facts.
Page 4, lines without numbers: the “nth” should looks probably like “n-th” not “n to the power of t, multiplied by h”.
The “n-th” issue has been fixed.
Page 4, lines 143-145. The sentence means exactly the same as described above in unnumbered paragraph.
We shortened the definition and removed redundant facts.
Page 5, description of Figure 1 – the number 6, which is marked on the Figure, is missing in the description.
We added the number 6 in the description of the Figure 1.
Page 5, line 170. Number not amount – amount is for uncountable nouns.
We changed “amount” with “number” in line 170.
Page 6 (not only) Algorithm 1. I strongly suggest to use the same letters for the number of samples, features etc. in whole article. For example – Algorithm 1 – M means number of features then, Table 2 (page 7) – the number of features is “m”. Number of samples is “n” or “N”. Take a look on your equations, Figure 1, Algorithm 1 and tables and use the same letters, please.
We uniformed the notation of samples and features in Equation 2 and 3, and Algorithm 1.
Page 7, I suggest to read again the first three sentences of section 3. In my opinion the information there are doubled – i.e. the “definition” of OpenGDC.
We shortened and improved the description of OpenGDC.
Page 7, line 209. The section number at the end of the sentence is missing.
We fixed the section number in line 209.
Page 7, line 212. Not “amount” of samples but the “number” of samples. Amount is for uncountable nouns.
We changed “amount” with “number” in line 212.
Page 7, line 215. What for is “i” at the end of the sentence? Should it be italic?
We simplified the sentence in line 215.
BigBioCl or BIGBIOCL or BIGBIOGL? All of them are in the article, but only one is correct.
We uniformed the name of the BIGBIOCL tool.
Round 2
Reviewer 1 Report
The authors addressed all my comments in an appropriate manner. I do not have any further objections except for maybe the following:
1) Improvement of English
2) Further improvement on making the text concise